# Low-Dose Empagliflozin Improves Systolic Heart Function after Myocardial Infarction in Rats: Regulation of MMP9, NHE1, and SERCA2a

**DOI:** 10.3390/ijms22115437

**Published:** 2021-05-21

**Authors:** Jana Goerg, Manuela Sommerfeld, Bettina Greiner, Dilyara Lauer, Yasemin Seckin, Alexander Kulikov, Dmitry Ivkin, Ulrich Kintscher, Sergey Okovityi, Elena Kaschina

**Affiliations:** 1Charité-Universitätsmedizin Berlin, Corporate Member of Freie Universität Berlin and Humboldt-Universität zu Berlin, Institute of Pharmacology, Center for Cardiovascular Research (CCR), 10115 Berlin, Germany; jana-catherine.goerg@charite.de (J.G.); manuela.sommerfeld@charite.de (M.S.); bettina.greiner@charite.de (B.G.); dilyara.lauer@nuvisan.com (D.L.); yaseminseckin1992@gmx.de (Y.S.); ulrich.kintscher@charite.de (U.K.); 2DZHK (German Centre for Cardiovascular Research), Partner Site Berlin, 10115 Berlin, Germany; 3Department of Biotechnology, University of Applied Science, 13353 Berlin, Germany; 4Pavlov First Saint-Petersburg State Medical University, 197022 Saint Petersburg, Russia; fd1med@mail.ru; 5Saint-Petersburg State Chemical-Pharmaceutical University, 197376 Saint Petersburg, Russia; dmitry.ivkin@pharminnotech.com (D.I.); Sergey.Okovity@pharminnotech.com (S.O.)

**Keywords:** empagliflozin, myocardial infarction, MMP9, NHE1, SERCA2a

## Abstract

The effects of the selective sodium-glucose cotransporter 2 (SGLT2) inhibitor empagliflozin in low dose on cardiac function were investigated in normoglycemic rats. Cardiac parameters were measured by intracardiac catheterization 30 min after intravenous application of empagliflozin to healthy animals. Empagliflozin increased the ventricular systolic pressure, mean pressure, and the max dP/dt (*p* < 0.05). Similarly, treatment with empagliflozin (1 mg/kg, p.o.) for one week increased the cardiac output, stroke volume, and fractional shortening (*p* < 0.05). Myocardial infarction (MI) was induced by ligation of the left coronary artery. On day 7 post MI, empagliflozin (1 mg/kg, p.o.) improved the systolic heart function as shown by the global longitudinal strain (−21.0 ± 1.1% vs. −16.6 ± 0.7% in vehicle; *p* < 0.05). In peri-infarct tissues, empagliflozin decreased the protein expression of matrix metalloproteinase 9 (MMP9) and favorably regulated the cardiac transporters sarco/endoplasmic reticulum Ca^2+^-ATPase (SERCA2a) and sodium hydrogen exchanger 1 (NHE1). In H9c2 cardiac cells, empagliflozin decreased the MMP2,9 activity and prevented apoptosis. Empagliflozin did not alter the arterial stiffness, blood pressure, markers of fibrosis, and necroptosis. Altogether, short-term treatment with low-dose empagliflozin increased the cardiac contractility in normoglycemic rats and improved the systolic heart function in the early phase after MI. These effects are attributed to a down-regulation of MMP9 and NHE1, and an up-regulation of SERCA2a. This study is of clinical importance because it suggests that a low-dose treatment option with empagliflozin may improve cardiovascular outcomes post-MI. Down-regulation of MMPs could be relevant to many remodeling processes including cancer disease.

## 1. Introduction

Recent clinical studies demonstrated favorable cardiovascular effects of the antidiabetic drugs from the sodium-glucose cotransporter 2 (SGLT2) inhibitor class, including a reduction of cardiovascular death, non-fatal myocardial infarction (MI), heart failure, and non-fatal stroke, as well as all-cause mortality [1,2,3,4,5,6]. In experimental models, cardiac contractility was improved in heart failure with preserved and reduced ejection fractions [7,8,9,10,11], in ischemia/reperfusion [12], MI models [13,14,15,16,17], and diabetic cardiomyopathy [18]. Recently, the antiarrhythmic properties of SGLT2 inhibitors were shown in the ischemia-reperfusion model [19] and atrial fibrillation [20].

The glucose-lowering effect of SGLT2 inhibitors is a result of reduced glucose reabsorption from the primary urine due to the SGLT2 cotransporter inhibition in the proximal tubule of the kidney [21]. However, the exact cardio-protective mechanism of action is still unclear—especially since SGLT2 receptors are not expressed in the heart [22,23]. Several favorable effects of SGLT2 inhibitors such as increased diuresis [24], decreased arterial stiffness [25], weight and blood pressure reduction [1,26], and other cardiac benefits [27] contribute to positive outcomes in heart failure. The cardioprotective effects were also attributed to the modification of the cardiac metabolome and anti-oxidants [13], increased energy production from glucose, ketone bodies and fatty acid oxidation [10,28], anti-inflammation [29], and improved myocardial oxidative phosphorylation [30]. Moreover, empagliflozin reduced Ca^2+/^calmodulin-dependent kinase II (CaMKII)-activity and CaMKII-dependent SR Ca^2+^ leak in ventricular myocytes [31].

It has been hypothesized that the selective SGLT2 inhibitor empagliflozin (Empa) may directly interfere with the cardiac sodium hydrogen exchanger 1 (NHE1) [32]. Modulation of NHE1 activity by Empa, in turn, decreased the cytosolic sodium and calcium levels while increasing the myocytes mitochondrial calcium concentration [32]. In an isolated heart, a delayed ischemic contracture onset by Empa was associated with NHE1 inhibition [33].

Nevertheless, the regulation of NHE1 and related cardiac pumps such as sodium bicarbonate cotransporter 1 (NBCe1), Na^+^/Ca^2+^ exchanger (NCX), sarco/endoplasmic reticulum Ca^2+^-ATPase (SERCA)2a in cardiac tissues after MI have not been tested until now. SERCA2a is the primary cardiac isoform regulating intracellular Ca^2+^homeostasis. Its downregulation in heart failure leads to a loss of cardiac contractility whereas increased expression of SERCA2a improves contractility [34]. Down-regulation of SERCA2a after MI in rats was linked with diastolic dysfunction [35]. It was also assumed that a compensatory increase in NCX compensates for the reduced SERCA2a activity [34]. NCX is the main Ca^2+^ extrusion mechanism of the cardiac myocyte which is involved in the regulation of cytosolic Ca^2+^ concentration, repolarization, and contractility [36]. Its increased activity has been identified as a mechanism promoting heart failure, cardiac ischemia, and arrhythmia [36]. NBCe1 contributes to the control of intracellular pH in cardiomyocytes [37] and its blockade may reduce ischemic Na+ overload [38].

Moreover, a relationship between NHE1 and matrix metalloproteinase 9 (MMP9) might also exist in the heart; as well as neuronal tissues where MMP9 activity is dependent upon the expression and activation of NHE1 [39]. Therefore, Empa might contribute to the regulation of MMPs in the heart directly or via NHE1. We tested this hypothesis in an experimental MI model. We also designed the present study to assess the short-term effects of the treatment with the selective SGLT2 inhibitor Empa in normoglycemic rats, 1 week after MI. In contrast to previous experiments where the dose of Empa ranged from 20 to 30 mg/kg [14,15,40], we used a lower dose of 1 mg/kg. This dose was based on our preliminary cell culture experiments and was successful in our previous experimental heart failure study [11].

## 2. Results

### 2.1. Short-Term Hemodynamic Effects of Empagliflozin

In the control healthy rats, Empa (1 mg/kg, i.v. bolus) increased the left ventricular maximal pressure (110.2 ± 5.3 mmHg vs. 86.1 ± 7.1 mmHg; *p* < 0.05) and the mean pressure (53.8 ± 4.7 mmHg vs. 36.0 ± 4.8 mmHg; *p* < 0.05) as compared to vehicle (Figure 1A,B). In addition, the max dP/dt (5685.6 ± 545.9 vs. 8446.0 ± 742.5; *p* < 0.05; Figure 1C) and Tau (0.01056 ± 0.00109 vs. 0.01549 ± 0.0011; *p* < 0.01) were increased from the initial value to 30 min after Empa application. Moreover, dP/dt min was decreased by tendency. The heart rate slightly decreased 30 min after application in vehicle and Empa groups.

The aortic stiffness parameters: augmentation index, pulse pressure (Pp) (Figure 1D), and second systolic peak pressure (P2) were increased to a similar extent in Empa and vehicle groups.

### 2.2. Hemodynamic Evaluation One Week after Myocardial Infarction

Post infarct mortality was 35%. The animals died within the first 24 h after MI. Hemodynamic parameters obtained one week after MI/sham operation are presented in Figure 2 and Table 1. Mean EF, FS and GLS were equal in all treatment groups at the start of the treatment. One week after MI, Empa protected against the impairment of systolic heart function as demonstrated by an improved global longitudinal strain (GLS) (MI + Empa; −20.99 ± 3.21% vs. MI + Veh; −16.64 ± 1.61%; *p* < 0.05) compared to vehicle (Figure 2A). Ejection fraction slightly increased in treated group vs. vehicle (Figure 2B). The E/A ratio tended to decrease. Blood pressure, heart rate, and arterial augmentation index were not modified by the treatment. Sham-operated animals treated with Empa showed an increase in stroke volume (274.8 ± 27.6 µL vs. 209.5 ± 22.77 µL; *p* < 0.01), fractional shortening (21.21 ± 1.58% vs. 16.01 ± 3.54%; *p* < 0.05), and cardiac output (105.8 ± 7.9 mL/min vs 86.6 ± 10.2 mL/min; *p* < 0.05) in comparison to sham treated with vehicle.

One week post-MI, Empa treated MI animals had a lower weight gain compared to vehicle (12.0 ± 9.9 g vs. 32.0 ± 13.6 g; *p* < 0.05). Heart/body weight index increased both in vehicle MI (*p* < 0.05) and Empa MI group (*p* < 0.01). There was no significant difference between the groups in the blood and urine glucose levels. Empa increased the urine glucose excretion in the sham group only by tendency (data not shown).

### 2.3. Regulation of Cardiac Transporters

The protein expression of cardiac transporters (NHE1, NBC, NCX, and SERCA2a) was examined in the peri-infarct zone of the left ventricle (Figure 3A,B). NHE1 was significantly down-regulated (2-fold, *p* < 0.05) by Empa post-MI compared to MI + vehicle, and decreased by the tendency in the Empa treated sham group compared to sham (1.4-fold). In the heart, NHE1 was localized at the plasma membrane of myocytes and transverse tubules as shown in the histological section of a rat heart (Figure 4A). Empa treated rats expressed less NHE1 compared to vehicle (Figure 4A(b)). In addition, in H9c2 cell line, Empa (5 µM) down-regulated NHE1 (110 kDa) (Figure 5).

The expression of NCX was increased after MI by tendency but not affected by Empa (Figure 3A,B). NBC1 (130 kDa) was not changed, whereas the 60 kDa form was upregulated (1.5-fold, *p* < 0.05) in the MI Empa group. SERCA2a was decreased (1.4-fold) post-MI, and its decrease was ameliorated by Empa (*p* < 0.05). After i.v. application of Empa in short-term experiments, cardiac expression of SERCA2a was 1.2-fold lower compared to the NaCl group. Immunohistological stain showed that SERCA2a was highly expressed in healthy cardiomyocytes co-localizing with sarcomere structures (Figure 4B(a)). Less SERCA2a staining was found in the MI group (Figure 4B(b)) and more in the Empa group (Figure 4B(c)).

### 2.4. Regulation of MMP2, MMP9, and TIMP1

In the left ventricle, an up-regulation of MMP9 post-MI (1.6-fold) was attenuated by Empa (*p* < 0.001) (Figure 3A,B). TIMP1 was not significantly regulated. Accordingly, the MMP9/TIMP1 ratio was regulated similarly to MMP9 (Figure 3B). In the peri-infarct zone, MMP9 was localized inside cardiomyocytes (Figure 4C(d)); around the nucleus (Figure 4C(e)); and co-localized with the inflammatory cells (Figure 4C(b,e)) and intramyocardial adipocytes. Less expression was found after Empa treatment (Figure 4C(c)).

In cardiac cell line H9c2, Empa decreased the activity of IL1α stimulated proMMP2, MMP2, and MMP9 compared to IL1α control (Figure 5a,b). The protein expression of MMP9 was also diminished by Empa (1–10 µM), although it was increased at a concentration of 50 µM (Figure 5). At the concentration of 1 µM, Empa prevented apoptosis, decreasing the apoptosis ratio by 25%.

### 2.5. Regulation of Fibrosis and Necroptosis

Post-MI increase of fibrotic markers TGF-beta1 (1.3-fold; *p* < 0.05) and smad2 (1.3-fold) expression was not modified by treatment (Figure 3A,B). MLKL, a marker of necroptosis, was also not regulated post-MI, although it was slightly decreased (1.4-fold) in Empa treated sham animals compared with sham (Figure 3A,B).

## 3. Discussion

SGLT2 inhibitors earned great attention from investigators due to the recently discovered pleiotropic cardioprotective effects. Besides favorable actions on the heart by treating diabetes [1,26], SGLT2 inhibitors also protected normoglycemic animals in heart failure models [7,8,10,11], myocardial infarction [13,14], and occlusion/reperfusion models [12]. Nevertheless, in clinical [4,5] and experimental [7,8,14] studies, there are controversial issues regarding systolic and diastolic heart function improvement. The mechanism of cardiac action is also not fully understood.

Our study showed that short-time application of Empa to healthy animals increased the ventricular systolic pressure, mean pressure, the max dP/dt, and maximal pressure in the aorta. Similarly, the application of Empa in a low dose in the sham group showed an increase in cardiac output, stroke volume, and fractional shortening. Thus, Empa improved the systolic heart function in non-diseased rats beyond modifying the heart rhythm.

Reduced global longitudinal strain (GLS) post-MI points to the favorable effect of Empa on the systolic function by early left ventricular remodeling. We could not demonstrate an improvement of the diastolic function; however, the diastolic parameter E/A tended to decrease after treatment.

Empa did not influence vascular parameters such as arterial augmentation index, maximal pressure in aorta (P2), and pulse pressure (Pp). Therefore, its favorable hemodynamic effects could be primarily attributed to improved cardiac contractility. These findings on the systolic function are in agreement with several recent studies [7,8,10,14].

An increase in the indices of force and the rate of change in variables of ventricular contraction (fractional shortening, stroke volume, and cardiac output) after Empa administration suggests a positive inotropic effect. Thus, increased contractility may reduce a mismatch between oxygen delivery to tissue and organ demand by cardiovascular pathology.

Blood pressure was not lowered by the treatment, and rather tended to increase in short-term experiments. This result is in agreement with a heart failure study in non-diabetic rats by Connelly et al. [15].

One-week treatment with a low dose of Empa induced a markable decrease in weight. The sham-treated group also showed the same results. This effect, probably due to the catabolic state [14,15] could be unfavorable by non-obese or post-MI subjects.

Positive inotropy is generally attributed to the cardiac actin-myosin cycle that depends on intracellular cAMP, calcium transients, or calcium-myosin activation. In this study, we focused on calcium transporters. Recently, Baartscheer et al. [32] discovered that the NHE1 blockade by Empa, independent from the SGLT2 receptor, decreased cytosolic sodium and calcium levels; and increased myocytes mitochondrial calcium concentration. Here, we provide in vivo evidence that Empa ameliorates an up-regulation of NHE1 expression in the left ventricular post-MI.

Nevertheless, we did not find a subsequent down-regulation of the NCX which is also an essential regulator of cardiac contractility independent of sarcoplasmic reticulum Ca^2+^ load [36]. Its increased activity promotes heart failure, cardiac ischemia, and arrhythmia [36].

Since sodium bicarbonate cotransporter 1 (NBCe1) also contributes to the control of intracellular pH in cardiomyocytes, we were interested in its regulation. The NBC membrane family proteins are responsible for 30% of Na+ influx into the cells during the recovery from acidosis [37]. The increase in Na+ is crucial for the heart because it decreases the driving force of the NCX leading to Ca^2+^ overload. Furthermore, NBC blockade reduces ischemic Na+ overload in isolated rat hearts [38]. In our study, in heart tissues, the NBC 130 kDa band that corresponds to the NBCe1 transporter was not regulated, although the 60 kDa form was up-regulated. Unfortunately, the results obtained in cell culture experiments as well in the other tissues provided differential expression muster. Therefore, further investigations on NBCe1 regulation by using another experimental approach would be required.

Current evidence suggests a cooperative action of NHE1 as an ion transporter and as a membrane scaffold in promoting the assembly of signaling complexes [41]. Interestingly, we found a down-regulation of MMP9 protein expression in the heart by Empa. This effect on MMP9 was confirmed in cultured cardiac cells in which Empa decreased the MMP9 activity. The expression of MMP9 is probably regulated by NHE1 while it creates an acidic environment that, in turn, activates proteolytic enzymes. This suggestion is in agreement with the study by Putney and Barber [42] who found that loss of NHE1 activity in mammalian fibroblasts decreased the MMP9 of the cells. Similarly, MMP9 processing was altered in cells expressing a defective NHE1 [39]. Since the expression of a native MMP9 inhibitor TIMP1, as well as TGF-beta1/smad2 regulation, were not modified in the present study, their impact in MMP9 regulation by Empa could be excluded. Notably, down-regulation of MMPs by Empa is of major importance because this effect could be relevant to other diseases including cancer.

Sarco/endoplasmic reticulum Ca^2+^-ATPase (SERCA)2a is the primary cardiac isoform regulating intracellular Ca^2+^ homeostasis. Its increased expression improves myocardial contractility and Ca^2+^ handling [43]. Importantly, high mechanical stress is directly linked to SERCA2a down-regulation and slowing of relaxation [35]. In the post-MI model of the rat, regional diastolic dysfunction was linked to elevated wall stress adjacent to the infarction, resulting in down-regulation of SERCA, disrupted diastolic Ca^2+^ handling, and local slowing of relaxation [35].

In our study, SERCA2a expression in the left ventricular tissues adjusted to the scar was also decreased post-MI and its decrease was ameliorated by Empa. Notably, an increase in SERCA2a was accompanied by the down-regulation of MMP9. An interplay between SERCA2a and MMP9 has been previously shown in cardiomyocytes where ablation of MMP9 prevented contractile dysfunction by increasing SERCA2a and calcium transients [44]. Therefore, the favorable effect of Empa on cardiac contractility post-MI could be explained by the increased SERCA2a, where regulation seems to be interrelated with MMP9 and NHE1.

Therefore, Empa in low dose increases cardiac contractility in normoglycemic healthy rats and improves systolic heart function post-MI. These effects may be attributed to the up-regulation of SERCA2a and down-regulation of MMP9 and NHE1.

## 4. Materials and Methods

### 4.1. Ethics Statement

This study was carried out in strict accordance with the national and European guidelines for animal experiments with approval by the ethics commission of the regulatory authorities of the City of Berlin, Germany, the “Landesamt fuer Gesundheit und Soziales” (registration number G0128/17, 22 August 2017).

### 4.2. Animals

Male normotensive Wistar rats (6 weeks old; weight, 200 to 220 g; Charles River Laboratories Germany GmbH, Sulzfeld, Germany) were kept in an SPF (specific pathogen-free) barrier under standardized conditions with respect to temperature and humidity and were housed on a 12 h light/12 h dark cycle in groups of four animals with freely accessible food and water.

### 4.3. Experimental Protocol

The acute effect of Empa (1 mg/kg, in 0.5 mL NaCl, i.v. in the vena femoralis) on left ventricular (LV) pressure was examined in healthy animals under 1.5–2% isoflurane. The ventricular pressure was recorded over a time period of 30 min.

MI was induced by ligation of the proximal left anterior descending coronary artery as previously described [33]. The rats were randomly assigned to the following groups: Empa treatment (1 mg/kg daily p.o.) (n = 7), vehicle treatment (0.9% NaCl) (n = 8), and sham-operated controls (n = 6). Treatment was started 24 h after MI. Animals with ejection fraction <35% were excluded from the experiment before assignment to groups.

### 4.4. Analysis of Hemodynamic Parameters

Transthoracic Doppler echocardiography was performed preoperatively and on day 7 postoperatively in rats anesthetized with isoflurane (2%) with the use of the high-resolution imaging system Vevo 3100 (VisualSonics, Toronto, ON, Canada). Echocardiographic parameters are known to correlate with the heart structure and metabolic changes in rats [45].

The short axis and long axis were recorded in B- and M-Mode. The flow velocity of the mitral valve was calculated via Pulse Wave Doppler presented in 4-chamber view. All data were evaluated using the VevoLab software (Version 2.2.0, VisualSonics Inc., Toronto, ON, Canada) as previously described [46,47].

The invasive hemodynamic assessment was performed at the end of the study on day 7 using a fiber-optic pressure transducer catheter (Samba Sensors, Västra Frölunda, Sweden) and Chart5 software (Blood Pressure Module) for analysis as previously described [46]. Briefly, after anesthesia with isoflurane (2%), the catheter was inserted into the right carotid artery. After recording blood pressure and heart rate in the ascending aorta, the catheter was advanced into the left ventricle and the pressure-time indices (*dP/dt_min_* and *dP/dt_max_*) were recorded. Pulse wave analysis (PWA) was performed as previously described [48]. Diastolic pressure (Pd), pulse pressure (Pp), and systolic pressure (Ps) were determined and averaged on the central aortic pressure waveforms from at least 20 cardiac cycles.

### 4.5. Glucose Levels

Glucose levels were determined in rat serum and urine collected 7 days after MI by enzymatic photometric test using Glucose GOD FS kit (DiaSys Diagnostic Systems GmbH, Number 125509910021). Determination of glucose was based upon the Trinder technique, an oxidation reaction by glucose oxidase. Chinonimin was used as colorimetric indicator.

### 4.6. Immunoblotting Analysis

Protein preparation and Western blot analysis were carried out as described previously [49]. Immunoblotting was performed using NHE1 (Abbiotec, San Diego, CA, USA), NBC1 (Cohesion Bioscience, London, UK), NCX (Swant, Marly, Switzerland), SERCA2a (Dianova, Hamburg, Germany), MMP9 and GAPDH Ab (Abcam, Hiddenhausen, Germany), TIMP-1, TGF-beta1, Smad2 Ab (Santa Cruz Biotechnology Inc., Heidelberg, Germany), MLKL (Millipore, Merk, Darmstadt, Germany). GAPDH was used as a loading control. The antibodies dilution is presented in Appendix A. Immunoreactive bands were visualized by enhanced chemiluminescence (Amersham-Pharmacia, Freiburg, Germany) and quantified by densitometry with Scion Image software.

### 4.7. Immunohistochemistry

Paraffin-embedded cross-sections of the heart (4 µm) were stained and analyzed by quantitative morphometry (Biorevo BZ-9000, Keyence, Japan). Immunohistochemistry was performed using the avidin–biotin complex method according to the manufacturer’s instructions (Vectastain ABC, Vector Laboratories, Burligame, CA, USA). Peroxidase activity was visualized by 3-amino-9-ethylcarbazole (AEC, Vector Laboratories, Burlingame, CA, USA) or 3,3’-diaminbenzidin (DAB) (Sigma-Aldrich Chemie, Darmstadt, Germany). The following primary antibodies were used: NHE1, MMP9, and SERCA2a. The dilution of the first antibody was 1:200 incubated overnight at 4 °C. The second antibody was diluted 1:500 and incubated for 1 h at room temperature. The ABC reagent was prepared according to the protocol and given to the slides for 30 min. It binds with the second biotinylated antibody. The AEC substrate or the DAB combines with this complex and a color reaction occurs.

### 4.8. Cell Culture

Neonatal rat cardiomyocytes (H9c2 cell line, Sigma-Aldrich, Merk, Germany) were incubated for 10 h in high-glucose Dulbecco’s modified Eagle’s medium plus 10% fetal bovine serum and treated with 1 ng/mL recombinant interleukin 1α (IL1α) to induce cytokine expression and mimic the post-MI inflammatory response in the heart. Incubation was performed with or without co-incubation with 5, 10, 50 µM empagliflozin (Toronto Research Chemicals, Toronto, ON, Canada). Protein expression of NHE1 was evaluated by immunoblotting, and activities of MMP2 and MMP9 were analyzed with gelatin zymography as previously described [50].

### 4.9. Statistical Analysis

Results are expressed as mean ± SD. Multiple comparisons were analyzed using two-way repeated measures analysis of variance (ANOVA) followed by Bonferroni’s multiple comparisons test. Two-group comparisons were analyzed by the 2-tailed Student unpaired *t-*test for independent samples. Differences were considered statistically significant at the value of *p* < 0.05. Statistical analysis was performed using GraphPad Prism 6 (GraphPad Software Inc., La Jolla, CA, USA).

## 5. Conclusions

In conclusion, short-term treatment with Empa in a low dose increased cardiac contractility in normoglycemic rats and improved systolic heart function in the early phase after myocardial infarction.

Empa administration did not alter arterial stiffness, blood pressure, markers of fibrosis, and necroptosis.

Cellular mechanisms that minimize cardiac damage by Empa include anti-proteolysis via the inhibition of MMP9, which is associated with decreased expression of NHE1, and prevention of apoptotic cell death.

Our study is of clinical relevance because it suggests that a low-dose treatment option with Empa may improve cardiovascular outcomes post-MI. Down-regulation of MMPs by Empa is of major importance because this effect could be relevant to many remodeling processes including cancer disease.

## Figures and Tables

**Figure 1 ijms-22-05437-f001:**
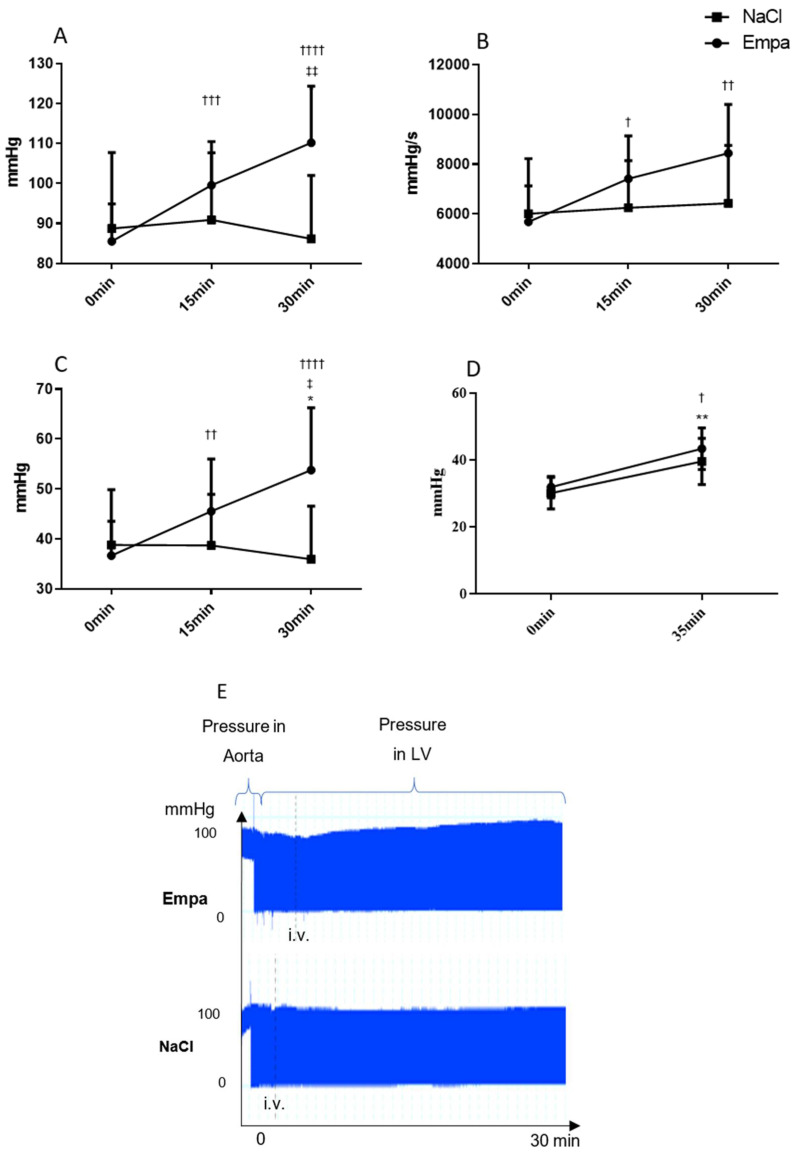
Hemodynamic parameters measured in the left ventricle by a cardiac catheter at baseline (0 min), 15 min, and 30 min after injecting NaCl or empagliflozin (Empa) (1 mg/kg/bw). Pressure mmHg (*x*-axis) at different time of measurements (*y*-axis). (**A**) Left ventricular maximal pressure (max pressure), * *P* < 0.05 Empa vs. NaCl 30 min; ††† *P* < 0.001, †††† *P* < 0.0001 vs. 0 min Empa, ‡‡ *P* < 0.01 vs. 15 min Empa. (B) Maximal rate of rise of left ventricular pressure (max dP/dt); † *P* < 0.05, †† P < 0.1 vs. 0 min Empa. (C) Mean Pressure; * *P* < 0.05 Empa 30 min vs. NaCl 30 min, †† *P* < 0.01, †††† *P* < 0.0001 vs. 0 min Empa, ‡ *P* < 0.05 vs. 15 min Empa. (D). Pulse pressure † *P* < 0.05, vs. 0 min Empa; ** *P* < 0.05 vs. 0 min NaCl. (E) Aortal and left ventricular pressure traces after Empa or NaCl administration, mean ± SD.

**Figure 2 ijms-22-05437-f002:**
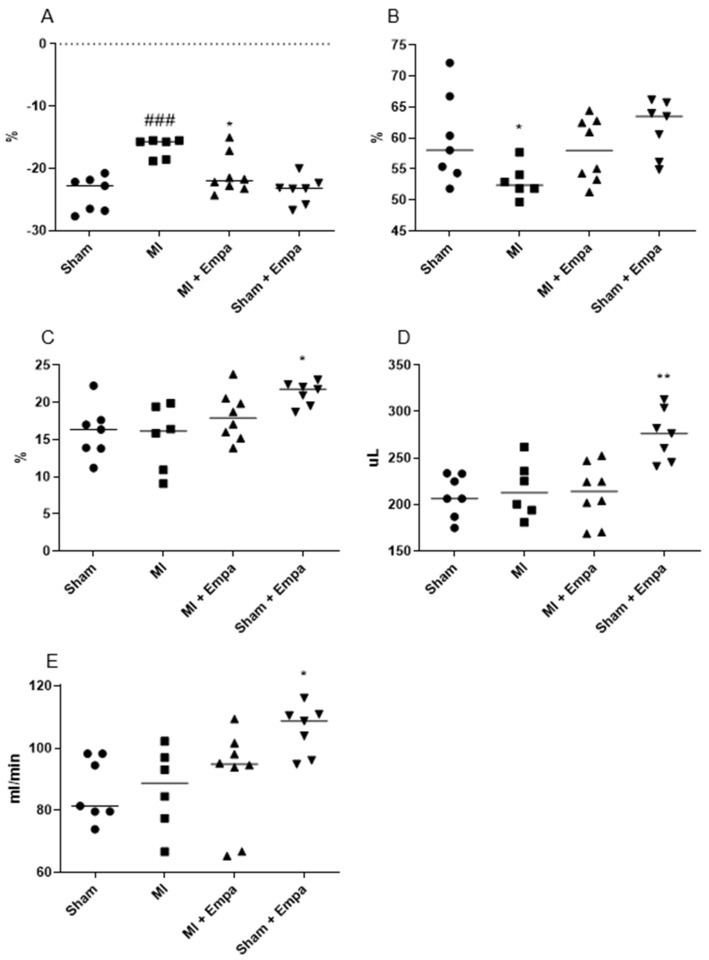
Hemodynamic parameters measured by Transthoracic Doppler Echocardiography 7 days after myocardial infarction (MI). (**A**) Global longitudinal strain (GLS); * *P* < 0.05 vs. MI, ### *P* < 0.001 vs. sham; 2-way Anova. (**B**) Ejection fraction (EF); * *P* < 0.05 vs. MI; unpaired t-test. (**C**) Fractional shortening (FS); *y*-axis: sham operation, MI vehicle, MI treated with Empa, sham operation treated with Empa; * *P* < 0.05 vs. sham; 2-way Anova. (**D**) Stroke volume (SV); ** *P* < 0.01 vs. sham, 2-way Anova. (**E**) Cardiac output; * *P* < 0.05 vs sham; 2-way Anova. (*n* = 7–8), mean ± SEM.

**Figure 3 ijms-22-05437-f003:**
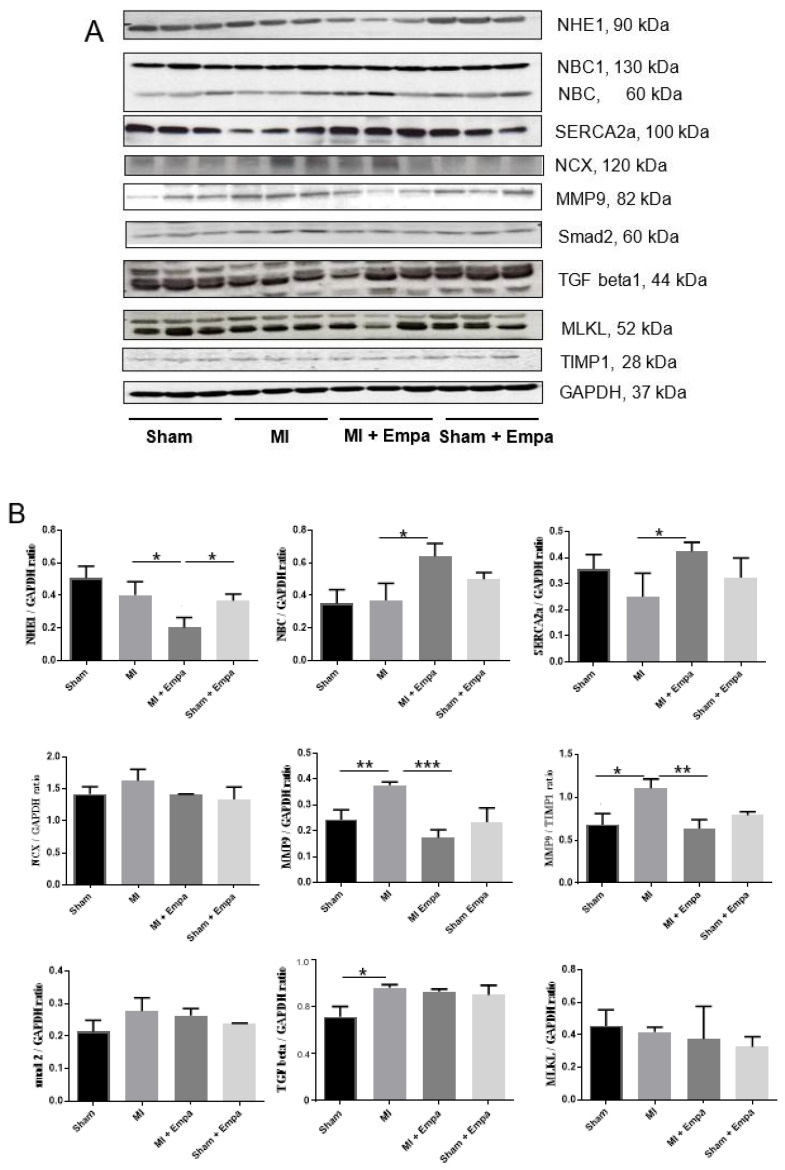
Molecular biological analysis of left ventricular 7 days after myocardial infarction. Western blot analysis of cardiac transporters (NHE1, NBC, NCX, SERCA2a), matrix metalloprotease 9 (MMP9), markers of fibrosis (smad, TGF-β1), marker of necroptosis (MLKL). (**A**) Representative Western blots: sham operation, MI vehicle, MI treated with Empa, sham operation treated with Empa. (**B**) Densitometric data of proteins are mean ± SEM (bars); * *P* < 0.05; ** *P* < 0.01; *** *P* < 0.001; two-tailed unpaired *t*-test.

**Figure 4 ijms-22-05437-f004:**
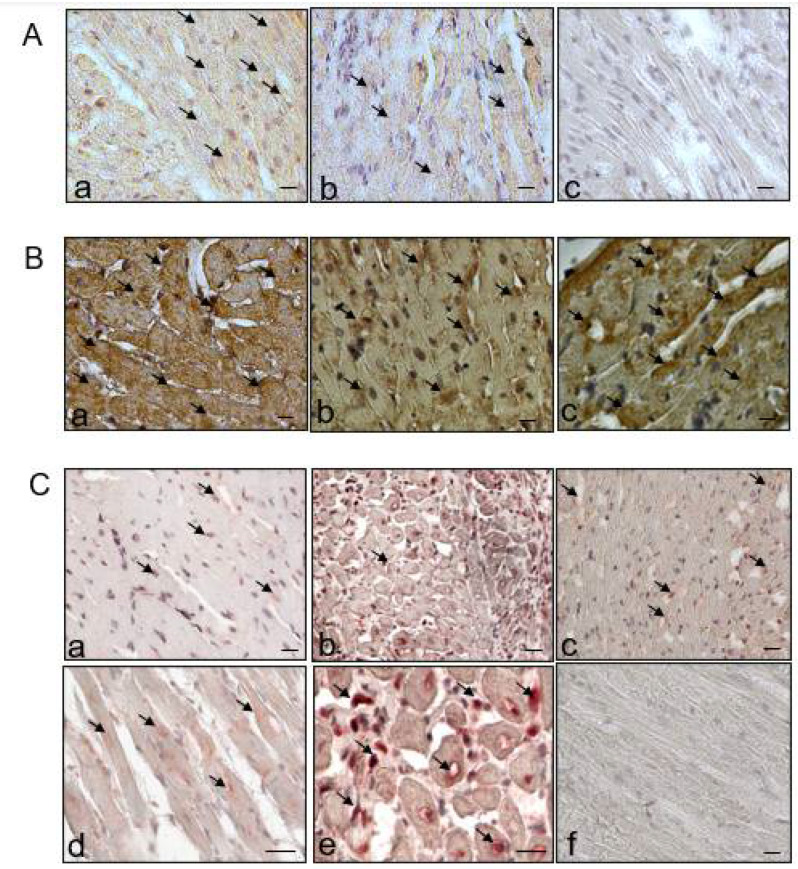
Cross-sections of the heart left ventricular, peri-infarct zone, immunohistological staining. (**A**) NHE1 (red stain), ×40; (**a**) MI vehicle; (**b**) MI treated with Empa; (**c**) negative control. (**B**) SERCA2a (brown stain), ×60; (**a**) sham operation; (**b**) MI vehicle; (**c**). MI treated with Empa. (**C**) MMP9 (red stain); (**a**) sham operation ×40; (**b**) MI vehicle, ×40; (**c**) MI treated with Empa, ×40 (**d**) MI vehicle ×60, MMP9 expression within cardiac myocytes; (**e**) MI vehicle ×60, MMP9 expression around the inflammatory cells and nuclei; (**f**) negative control. Scale 100 pixels.

**Figure 5 ijms-22-05437-f005:**
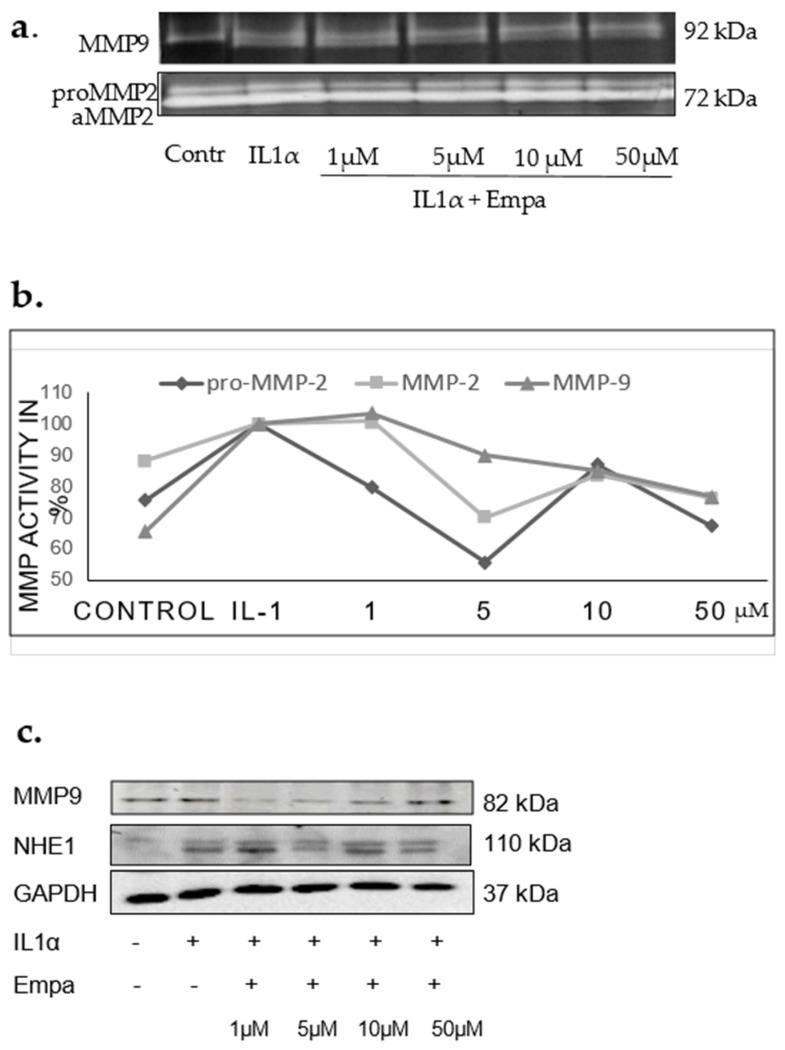
(**a**) Gelatin zymography, H9c2 cells, representative zymogram (**b**). Results from 3 experiments, 3 replicates; proMMP2, MMP2, MMP9 activity related to control after stimulation with IL1α and treatment with Empa 1, 5, 10, 50 µM. (**c**) Representative Western blots of MMP9, NHE1, GAPDH expression in H9c2 cells, 48 h after stimulation with IL1α.

**Table 1 ijms-22-05437-t001:** Hemodynamic parameters measured 7 days after myocardial infarction via echocardiography.

	Sham	Sham + Empa	MI + Vehicle	MI + Empa
Heart rate (bpm)	413 ± 21	386 ± 14	401 ± 19	426 ± 34
EF (%)	59.9 ± 7.2	61.6 ± 4.5	53.0 ± 2.7	58.0 ± 5.1
GLS (%)	−24.1 ± 2.8	−23.5 ± 2.2	−16.6 ± 1.6 †††	−21.9 ± 3.2 *
FS (%)	16.0 ± 3.5	21.2 ± 1.6 †	15.3 ± 4.4	18.1 ± 3.2
SV (uL)	209.5 ± 22.8	274.8 ± 27.6 ††	216.4 ± 30.2	211.7 ± 31.5
CO (mL/min)	86.5 ± 10.2	105.8 ± 7.9 †	86.8 ± 13.3	90.6 ± 16
LVIDd (mm)	6.6 ± 0.3	7.3 ± 0.2	6.9 ± 0.5	6.9 ± 0.6
LVIDs (mm)	3.8 ± 0.6	3.8 ± 0.6	3.6 ± 0.7	3.7 ± 0.9
E/A Ratio	1.71 ± 0.24	1.58 ± 0.14	1.84 ± 0.31	1.69 ± 0.25
IVRT (mm/s)	0.05120 ± 0.00743	0.05123 ± 0.00508	0.05715 ± 0.00841	0.04585 ± 0.00810
IVCT (mm/s)	0.04754 ± 0.01319	0.04417 ± 0.01057	0.04616 ± 0.00893	0.03804 ± 0.00965

* *P* < 0.05 vs. vehicle treated animals; † *p* < 0.05 vs. sham, †† *p* < 0.01 vs. sham, ††† *p* < 0.001 vs. sham. Heart rate (bpm—beats per minute). EF—ejection fraction; GLS—global longitudinal strain; FS—fractional shortening; SV—stroke volume; CO—cardiac output; LVIDd—end-diastolic left ventricular internal diameter; LVIDs—end-systolic left ventricular internal diameter; E/A ratio—ratio of early to late mitral inflow diastolic velocities. IVRT—isovolumic relaxation time; IVCT—isovolumic contraction time (*n* = 6–8), mean ± SEM.

## Data Availability

The data presented in this study are available on request from the corresponding author.

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
