# Peer review of "Low-Dose Empagliflozin Improves Systolic Heart Function after Myocardial Infarction in Rats: Regulation of MMP9, NHE1, and SERCA2a"

_ijms, 2021, doi:10.3390/ijms22115437_

Round 1

Reviewer 1 Report

Dear authors,

Please improve your introduction with more data and references

Please tell us more about ethics of the study (committee number)

Please add more conclusions 

I think you can add the following reference VALIDATION AND CHARACTERIZATION OF A HEART FAILURE ANIMAL MODEL, 

FARMACIA,Volume: 64, Issue: 3, Pages: 435-443

Best regards

Author Response

Dear Reviewer,

Thank you for the carefull review of our manuscript. Please find attached our response.

Sincerely,

Elena Kaschina

Reviewer 2 Report

The manuscript "Low-Dose Empagliflozin Improves Systolic Heart Function after Myocardial Infarction in Rats: Regulation of MMP9, NHE1 and SERCA2a" is an interesting study. But the discussion part needs improvement.

Comments

  1. In the abstract, the significance of the study is missing.
  2. Provide details of the x and y-axis in figure 1.
  3. In all figure-legends explain how the data was expressed (mean±SD or SE)
  4. Include heart rate details of animals in Table 1.
  5. Y-axis details for figure 2 are missing.
  6. In discussion, provide more recent references for explaining cardiac transporters (NHE1, NBC, NCX, SERCA2a).
  7. Explain the mechanistic cause of increased FS, SV, and CO in Empa treated animals (Figure 2).
  8. TIMP1 WB images were missing in Figure 3A.
  9. TGF beta western blot images and quantification data look different, redo quantification. In the figure legends, explain how the data was expressed (mean±SD or SE).
  10. Scalebar is missing in some images; why? Provide uniform images. Figure 4C image b, the small square box looks very confusing.
  11. Provide animal ethical approval details.
  12. Include more details of the Glucose GOD FS kit.
  13. What was the Ab dilution in immunoblotting?
  14. Provide details of kits used in immunohistochemistry.

Author Response

Dear Reviewer,

Thank you for carefull review of our manuskript. Please find attached our response.

Sincerely,

Elena Kaschina

Round 2

Reviewer 2 Report

The authors thoroughly modified the manuscript "Low-Dose Empagliflozin Improves Systolic Heart Function after Myocardial Infarction in Rats: Regulation of MMP9, NHE1, and SERCA2a" as per reviewers suggestion. Now it is acceptable for publication.

Minor spelling correction

Line 127 Check spelling beats per minute.

Author Response

Dear Reviewer,

Thank you for carefull evaluation of our manuscript. We are sorry for this error, and we have made correction  (indicated yellow).

Best regards

Sincerely,

Elena Kaschina